# Hepatitis E Virus in Domestic Ruminants and Virus Excretion in Milk—A Potential Source of Zoonotic HEV Infection

**DOI:** 10.3390/v16050684

**Published:** 2024-04-26

**Authors:** Gergana Zahmanova, Katerina Takova, Georgi L. Lukov, Anton Andonov

**Affiliations:** 1Department of Molecular Biology, University of Plovdiv, 4000 Plovdiv, Bulgaria; 2Department of Technology Transfer and IP Management, Center of Plant Systems Biology and Biotechnology, 4000 Plovdiv, Bulgaria; 3Faculty of Sciences, Brigham Young University–Hawaii, Laie, HI 96762, USA; 4Department of Medical Microbiology and Infectious Diseases, Max Rady College of Medicine, University of Manitoba, Winnipeg, MB R3T 2N2, Canada; anton.andonov@umr.umanitoba.ca

**Keywords:** hepatitis E virus (HEV), ruminants, HEV zoonotic transmission, HEV excretion, milk, foodborne infection, HEV cross-species infection

## Abstract

The hepatitis E virus is a serious health concern worldwide, with 20 million cases each year. Growing numbers of autochthonous HEV infections in industrialized nations are brought on via the zoonotic transmission of HEV genotypes 3 and 4. Pigs and wild boars are the main animal reservoirs of HEV and play the primary role in HEV transmission. Consumption of raw or undercooked pork meat and close contact with infected animals are the most common causes of hepatitis E infection in industrialized countries. However, during the past few years, mounting data describing HEV distribution has led experts to believe that additional animals, particularly domestic ruminant species (cow, goat, sheep, deer, buffalo, and yak), may also play a role in the spreading of HEV. Up to now, there have not been enough studies focused on HEV infections associated with animal milk and the impact that they could have on the epidemiology of HEV. This critical analysis discusses the role of domestic ruminants in zoonotic HEV transmissions. More specifically, we focus on concerns related to milk safety, the role of mixed farming in cross-species HEV infections, and what potential consequences these may have on public health.

## 1. Introduction

HEV is the most common cause of acute viral hepatitis worldwide with 3.3 million symptomatic cases annually, leading to ~70,000 deaths [1,2]. Globally, based on anti-HEV IgG positivity, one in eight people has had an HEV infection [3]. Usually, HEV infection causes an asymptomatic or self-limited disease, but in immunocompromised patients it can progress into chronic hepatitis or may affect other organs [4,5]. In the general population, the mortality rate ranges from 0.5 to 4%, while in pregnant women the mortality rate can be as high as 25% [6]. The majority of patients with HEV infections come from areas with few resources (Asia and Africa), where the infection is transmitted mainly via the faecal–oral route and due to the nonzoonotic HEV genotypes 1 and 2 (HEV-1 and HEV-2) [7]. Zoonotic transmission of HEV is the primary cause of the rising number of autochthonous cases of human infection in industrialized nations [8,9]. This occurs primarily via the consumption of undercooked or raw meat from infected pigs and wild boars, which are thought to be the main reservoirs of HEV [10,11]. The fact that human HEV seroprevalence has reached noteworthy levels in many countries without recorded outbreaks (42% in the UK, 22% in France, up to 30% in Portugal, and 21% in the USA), highlights the silent and unnoticed spreading of this virus that could lead to potentially surprising and life-threatening consequences for individuals or populations at risk [12,13,14].

HEV is a single-stranded RNA virus, whose genome is 7.2 kb long and has three open reading frames (ORFs), ORF1, ORF2, and ORF3, flanked by 5′ and 3′ untranslated regions (UTRs) [15,16]. HEV genotype 1 has an additional ORF4, whose translation stimulates viral replication [17]. ORF1 encodes the nonstructural polyprotein critical for viral replication [18]. ORF2 and ORF3 proteins are translated from a 2.2 kb bicistronic subgenomic RNA [19]. ORF2 encodes the viral capsid protein, ORF3 partially overlaps with ORF2 and is translated into a small 113–115 amino acids phosphoprotein with multiple functions [20,21]. The ORF3 protein has been discovered to be a functional ion channel protein (as a viroporin) and to be essential for virion release from infected cells [22]. Furthermore, the ORF3 protein serves as an adaptor to connect intracellular transduction pathways, to lower the host inflammatory response and shield virus-infected cells. The infected organisms produce two types of HEV virions: quasi-enveloped (eHEV) particles, which are wrapped in a lipid envelope, and nonenveloped particles (neHEV) [23]. The neHEV particles are secreted in faeces, while the eHEV particles are circulated in the blood and in vitro, via infected cell-culture supernatants [24]. The enveloped forms of the virus are somewhat more impenetrable to the anti-HEV antibody, because the host membrane completely masks the viral antigens [25]. eHEV is responsible for cell-to-cell spread within the host [26]. neHEV is characterized by a high initial cell attachment (~10-fold higher) than that of the enveloped virions [27].

HEV belongs to the family *Hepeviridae*, which contains two subfamilies (*Orthohepevirinae* and *Parahepevirinae*), five genera (*Avihepevirus*, Chirohepevirus, *Paslahepevirus*, *Rocahepevirus*, *Piscihepevirus*), and ten species (*A. magniiecur*, *A. egretti*, *C. eptesici*, *C. rhinolophi*, *C. desmoid*, *P. balayani*, *P. alci*, *R. ratti*, *R. eothenomi*, and *P. heenan*) [15].

HEV genotypes that can infect people belong to the species *Paslahepevirus balayani* and *Rocahepevirus ratti. P. balayani* includes eight genotypes (HEV-1 to HEV-8). HEV-1 and HEV-2 exclusively infect humans and are transmitted as waterborne infections in areas with poor sanitation conditions. HEV-1 strains are distributed in Asia and most countries in sub-Saharan Africa [28]. HEV-2 was identified during a hepatitis E outbreak in Mexico in 1986; additionally, several studies have reported that HEV-2 is distributed in Africa [29,30,31,32]. HEV-3 and HEV-4 are identified as zoonosis, infecting humans and many animal species [33]. HEV-5 and HEV-6 were observed among wild boars [34,35]. HEV-7 infects mostly dromedary camels; however, its zoonotic potential has been demonstrated as well, albeit in a single human case (an immunosuppressed patient from the Middle East, who consumed camel meat and milk) [36,37]. HEV-8 was identified among Bactrian camels in China [36]. Figure 1 presents *Paslahepevirus balayani* animal species distribution.

The autochthonous hepatitis E in Europe and the Americas is caused by HEV-3 [11,39,40]. HEV-4 is prevalent across Asia, including Japan, China, and Indonesia [41]. The primary animal reservoirs of HEV-3 and 4 are domestic pigs and wild boars, though deer has also been implicated in human transmission [42]. Anti-HEV IgG seropositivity in domestic pigs ranges from 30% to 100% [40,43,44,45,46,47,48,49,50,51]. A systematic review of HEV in wild boars stated that the pooled HEV-specific antibody seroprevalence in wild boar was 28% (CI_95%_ 23–34) [35]. HEV RNA is often detected in liver, muscle, and serum samples from pigs and wild boar with the HEV RNA prevalence 8% (CI_95%_ 6–10) [35]. In recent years, many studies have revealed that HEV-3 and HEV-4 can also infect domestic ruminants [52,53]. HEV-3 and HEV-4 were also detected in horses, donkeys, and mules [54,55]. 

HEV-3 also includes some unassigned strains, such as rabbit HEV (subtype 3ra) [56]. These strains belong to a separate clade within genotype 3 and share 73% to 80% nt identity with other HEV-3 subtypes, and its genotyping is still under consideration [34,56].

Rats are the primary reservoir of a recently identified rodent HEV (*Rocahepevirus ratti*, HEV-C1) generally referred to as rat HEV; however, this virus can also infect foxes and occasionally cause hepatitis in humans too [57]. In cats and dogs, HEV-3 and rat HEV-C1 can be detected; this supports the hypothesis that these animals are infected with HEV via consumption of infected food or via contact with the excrement of infected animals [58]. 

Of all the abovementioned animals, pigs, boars, and deer remain the primary zoonotic hosts, with rabbits, rats, and camels implicated on rare occasions. Recently, a growing awareness of the zoonotic potential of HEV in domesticated ruminants has risen, especially concerning the excretion of HEV in milk and the implications of a possible milk-borne transmission [59,60,61,62,63]. However, the information regarding the possibility of milk-borne transmission of the virus is either sparse or contradictory. This is why this critical analysis aims to provide an overview of the previous research that has found HEV in ruminants and their milk, and to discuss to what extent the transmission of the virus via milk is possible.

## 2. Hepatitis E Virus in Domestic Ruminants 

Most ruminants belong to the bovidae family (*Bovidae*), a subfamily of which, *Bovinae*, includes cattle, related yaks, and buffalo, among other, and another subfamily, *Caprinae*, comprises 20 species, including domestic goats and sheep [64]. According to the Food and Agriculture Organization of the United Nations, there are over one billion each of domesticated cattle, sheep, and goats in the world in addition to the global population of approximately one billion pigs. The latter are known to be a well-established reservoir and source of HEV infection; however, the current data for domestic ruminants requires more robust assessment of their role perhaps as an under-recognized infection threat.

### 2.1. Seroprevalence of Hepatitis E Virus in Ruminants 

The anti-HEV seroprevalence in ruminants varies widely in different research studies, depending on the endemicity of infection, different levels of economic developments of investigated countries and sensitivity/specificity of anti-HEV immunoglobulins kits from various manufacturers. Data on the HEV seroprevalence, the detected type of immunoglobulins, and the used detection kits were summarized in Appendix A.

Worldwide, the HEV seroprevalence in cows varies from 0.0% to 29.35%; in Europe, anti-HEV antibodies (Abs) were proven in only one study from Bulgaria (7.7%) [60], it was excluded in Spain (0.0%) [65]. In the Americas, data vary widely: the USA (20.4%) [66] and Brazil (1.42%) [67]. In Asia and Africa, data on anti-HEV seroprevalence are relatively high: India (6.9%) [68]; China (6.5%) [69], (18.7%) [70], (28.2%) [71], (29.35%) [72]; Turkey (16.5%) [73]; Jordan (14.5%) [74]; Egypt (21.6%) [75]; Burkina Faso (up to 26.4%) [76,77]. Meanwhile, South Korea [78] and Nigeria [79,80] had 0% seroprevalence. 

In Europe, a relatively higher HEV seroprevalence is observed in sheep and goats than in cattle. Seroprevalence in sheep varies from 0% to 24.4%. The highest seroprevalence was observed in Bulgaria (24.4%) [60]; followed by Italy (21.3%) [81] and (21.6%) [82]; Portugal (16.6%) [83]; and Spain (2.1%) [84] and (1.92%) [65]. In the Americas there has been only one study from Brazil, which showed 0% seropositivity [67]. In Asia, China is characterized with a higher seroprevalence of up to 35.2% [52,70,72,85,86]; while levels of 12.7% in Jordan [74] and 5% in Turkey were recorded [73]. In Africa the following levels were recorded: 10.5–31.8% in Nigeria [79,87], 12.0% in Burkina Faso [76], and 4.4% in Egypt [75].

Seroprevalence in goats varies from 0% to 100% [59,62,65,69,71,75,76,82,84,88,89,90,91]. In Europe, the highest seroprevalence was again observed in a study from Bulgaria (24.4%) [60], followed by Spain (up to 13.8%) [84] and Italy (11.4%) [82]. Meanwhile, the following levels were recorded elsewhere: the USA (16%) [90]; in Asia: China (0–46.7%) [59,69,71,88,92] and India (0–100%); and in Africa: Egypt (9.4%) [75] and Nigeria (0–32.7%) [79,80].

Although it is difficult to systematically analyse the reasons for the wide variation in seroprevalence, it seems prudent to suggest that HEV infection or that of an HEV-related virus is well established in domesticated ruminants. This is in agreement with numerous, not only serological, but also molecular studies in wild ruminants such as sika deer, red deer, roe deer, and other species belonging to the *Cervidae* family, which have been well characterized as hosts of HEV 3–4 genotypes [93,94,95]. One notable difference is the lack of, or the scarcity of, HEV RNA detection in domestic ruminants [66]. As mentioned above, this may be due to infection with HEV-like viruses, which are genetically quite different (and not detectable using PCR), but still induce a cross-reactive HEV immune response. Indeed, a highly genetically divergent HEV-like moose virus has been identified in Sweden [96]. Similar rationale could be extrapolated from a study of HEV in goats; conducting a detailed sampling of faeces and serum on a weekly basis from birth to 14 weeks old goats, HEV RNA detection was never detected, and at the same time, anti-HEV seroconversion has been observed [90].

### 2.2. HEV RNA Prevalence in Domestic Ruminants 

The presence of HEV viral RNA has been confirmed in various samples from ruminants. Silva et al. determined that the overall pooled prevalence of HEV RNA in ruminants was 0.02% (0.01–0.03, 95% CI) [97]. Whole genome sequence analysis of HEV isolates collected from ruminants showed a close phylogenetic relationship between the strains detected in these animals and those observed in humans [53,98,99]. Despite this, some recent findings support the idea that ruminants can be infected via close contact with diseased pigs or wild boars and their excrement, but they do not serve as a real viral reservoir for HEV [100]. Table 1 summarized the results for HEV RNA presence in cattle, goats, and sheep.

#### 2.2.1. HEV in Goats

Currently, the most compelling evidence of a domestic ruminant as a host of HEV infection has been identified in goats. Apart from the serological data mentioned above, there are credible reports of molecular corroboration (HEV RNA) in most discernible types of samples such as faeces, serum, and liver. An overall prevalence of 9.2% (11/119) for HEV RNA was detected in faeces of adult goats from six farms in Italy [101]. Phylogenetic analysis revealed that the goat HEV strains belonged to genotype 3c highly matching (94.2–99.4% sequence homology) to a wild-boar HEV strain previously identified in the same geographical area. A much higher prevalence of HEV active infection was reported in another study from China; HEV RNA was detected in 70.27% of faeces from 74 animals and in 53.57% in serum samples of 28 animals [62]. In this study, all HEV strains belonged to genotype 4h with high sequence homology (>99.6) to human, pig, and cow strains recovered from the same area. Although the numbers were very low, all milk samples tested were also positive for HEV RNA (4/4). It is worth noting that the viral loads in the faeces, serum, and milk were similar. 

HEV RNA was also detected in the livers of 2 out of 50 goats (4%) in a slaughterhouse in China [59]. This may seem like a low rate, but it is none the less meaningful in the light of the average findings of HEV in pig livers during slaughter (10–15%); however, this is age related and may vary significantly from <1% in livers of adult (>12 months) pigs to almost 40% in 3-month old ones [108,109,110,111,112].

Several other studies have found HEV RNA in goat milk, albeit in very low levels: 0.7% (2/280) in Egypt in goats living together with the owners in the same housing [113] and 1.4% (4/290) in the Czech republic [114]. A study in Turkey reported a much higher rate of HEV RNA detection (18.46%–12/65 samples); however, the data is open to debate as 8 of these belonged to HEV genotype 1, a highly unlikely finding given the fact that members of this genotype are strictly transmissible between humans [98]. It is important to mention that there is a growing number of published reports from diverse geographical regions with no molecular evidence of HEV infection (no HEV RNA detection) when testing was performed of the most relevant specimens such as serum, bile, and faeces [61,82,84,88,90,115,116].

#### 2.2.2. HEV in Sheep

The earliest reporting of HEV infection using molecular testing was from China in 2010; 11.1% (6/54) of faecal samples from a single sheep farm tested positive for HEV RNA using nested PCR with a sequence homology of 94–95% to HEV genotype 4 [103]. In another study from China, HEV-RNA was detected in the livers of slaughtered sheep in 4 out of 75 animals (5.3%) [52]. Interestingly the serum samples of the animals were negative for HEV RNA. A low prevalence of HEV RNA (2%) was observed in sheep faeces (4/200) in a study from Mongolia; however, the detection rate was higher in liver samples (3/60, or 5%) [102]. All HEV strains belonged to genotype 4.

The presence of HEV in Europe was reported in two studies from Italy (10.4% HEV RNA prevalence in 20 out of 194 faecal samples, but only 1.6% in serum) [81] and from the Czech Republic in milk samples (4/290, 1.4%) [114]. As was the case with goats above, no HEV RNA was found in equal number of studies in faecal, serum, and/or liver specimens from China, India, Turkey, and Spain [73,84,89].

#### 2.2.3. HEV in Bovines, Genus Bos with the Spotlight on Cow Milk

Most published reports on the existence of HEV in cattle originate from China; the first three describe a low prevalence of HEV RNA in serum samples, ranging from 0.19% (3/1692) to 0.8% (7/912) in cows and 3.15% (8/25) in yellow cattle [70,71,107]. However, a recent study described a very high HEV RNA detection rate in 140 faecal samples—37.14% [53]. Phylogenetic analysis revealed that the HEV strains in this study belonged to genotype 4 and shared more than 99% sequence homology with local human and/or pig strains, suggesting a possible common source of infection. There are two more published records from China outlining HEV presence in two other species of genus *Bos*: 3 out of 92 (3.26%) faecal samples of domestic yaks and 4.72% (5/106) serum samples from buffaloes [63,117]. HEV RNA was also detected in milk of the latter in 7.5% (3/40). All of the PCR positive amplicons were characterized as HEV genotype 4.

Apart from China, the only other country in East Asia where HEV in cows has been detected so far is South Korea: one out of hundred livers for sale in grocery stores had molecular evidence of HEV genotype 4 [105]. Further to the west, a study from Egypt [118] described the presence of HEV in 0.2% of milk samples (1/480); the HEV genotype was 3a. Nearly a third of milk samples from a study in Turkey turned out to be positive for HEV RNA—29.16% (14/48); however, only two of these belonged to the zoonotic genotypes 3a and 4c, with the rest being characterized as genotype 1 [98]. Geographically close to Turkey, Romania is the only European country with evidence of HEV infection in cows: 1 out of 49 (2.04%) stool samples were HEV RNA positive [106]; however, no genetic characterization was performed.

Advancing further westward, a study conducted in São Tomé and Príncipe, off the western equatorial coast of Central Africa, recorded a single cow stool sample positive for HEV RNA—14.29% (1/7), genotype not determined [61]. In the Americas, a single comprehensive study in Brazil analysed 240 liver samples from one of the largest cattle producing states and showed the presence of HEV infection in 5.4% in seven different cattle herds [119]. The sequencing of one isolate revealed that it belonged to genotype 3. 

In summary, most of the studies regarding HEV active infection in cows and milk, or for that matter in other domesticated ruminants as well, originate from China. However, this exploration of HEV in cattle will be somewhat restricted in scope without the negative data on the subject. Over a dozen recently reviewed studies failed to identify the presence of HEV infection, [97] including thousands of dairy milk, stool, serum, bile, and liver specimens from the United States of America (USA), Europe (Germany, Belgium, The Netherlands, Hungary, Croatia), and Asia.

Molecular analysis and epidemiological data on HEV circulation in ruminants are essential resources for monitoring the virus’ zoonotic spread and activating prophylactic measures putting into action.

## 3. Significance of HEV in Milk 

There is conclusive evidence in a growing number of reports demonstrating the presence of HEV in milk of domesticated ruminants, especially cattle, which were among the key domestic livestock animals in rural societies of the past as well as of current industrial agriculture [120]. Table 2 summarizes the data on HEV RNA in milk. 

The implications for potential HEV transmission are concerning; however, we must emphasize that no human cases via consumption of cow products, or for that matter of goat and sheep milk, have been described. Within this context, it is also important to mention that in the single reported case of a breastfeeding mother with acute HEV infection in whom HEV RNA was detected in both serum and milk (similar HEV RNA viral load), transmission of HEV was not observed in her 18-month-old toddler during breastfeeding [124]. Nevertheless, we have to acknowledge the worrisome possibility of potential transmission via cow milk raised by a single published report, showing that gavage of two rhesus macaques with HEV-contaminated cow milk, both raw and pasteurized, resulted in active infection [53]. Interestingly, while both viremia and HEV excretion in faeces were observed, no specific anti-HEV immune response was mounted upon follow up. Obviously, the currently available data is limited enough to obscure an underdiagnosed risk of HEV transmission via milk consumption. Alternatively, we can speculate that the risk is marginal, thus explaining the lack of empirical data on the subject. Although scanty, the current epidemiological data on HEV in ruminants, especially cattle, compared to that of pigs being the best-studied zoonotic reservoir of infection indicates that the low viral circulation in the former may be associated with a significantly diminished risk of human transmission. Indeed, most of the reviewed studies above suggest either very low prevalence rates of HEV in cow milk, or no evidence at all. Further, the observed low viral loads are less likely to be associated with viral transmission. 

At present, the contrast between the high prevalence of HEV RNA in cows in China compared to the low one, or the lack of it in other geographical regions such as Europe and the Americas, is explained on the basis of different breeding techniques; presumably, cows from traditional mixed domestic animal farms hosting diverse animals, including pigs, carry a higher risk of interspecies HEV transmission than in highly industrialized dairy farms where housing and management aim to prevent potential cross-species spread of infection [121].

The requirements for high-level hygiene standards may not be always properly followed in small, family-run farms with mixed animals; this may result in environmental HEV contamination of raw milk from bioaerosols and dust containing soilborne microbial organisms and faecal and animal skin microflora. The only other study outside of China that described a high prevalence of HEV in milk was conducted in Turkey [98]. The combined HEV RNA prevalence rate in cow, goat, and sheep milk was 19.1% (34/178); however, a remarkable 30 out of the 34 HEV positive samples belonged to genotype 1 and only four to the zoonotic genotype 3 and 4. Considering that HEV genotype 1 causes infection exclusively in humans, it is possible to hypothesize the environmental contamination of the milk due to the poor hygienic practices of the operators. Furthermore, if this is the case, one cannot exclude the same source of environmental contamination resulting in the detection of the zoonotic genotypes 3 and 4 in milk, instead of being attributed outright to an active HEV infection. Either way, a potential milk-borne transmission may ensue and justifies further research.

## 4. Milk and Milk Safety Measures to Prevent Foodborne Zoonotic HEV Transmission 

Due to the risk of foodborne HEV infection via the consumption of raw and undercooked meat products or raw milk, it is important to comprehend the thermostability of the HEV virion during food processing. One study proved that the pasteurization of milk does not inactivate HEV [53]. HEV-contaminated pork liver, incubated for 1 h at 56 °C, still contained intact HEV virions that could induce infection when pigs were inoculated with the liver extract [125]. Further, most studies showed that HEV cannot be inactivated via brief heating to 56 °C or 60 °C [126]. To prevent food-borne hepatitis E infection, the use of proper food processing conditions is essential. Several studies have shown that HEV can be effectively inactivated by cooking contaminated meat for 20 min at an internal temperature of 71 °C, or by boiling the contaminated milk (~100 °C) for several minutes [53,126]. It is also necessary to mention that HEV-4 has a higher thermal stability than HEV-3, and that HEV-4 is the most prevalent genotype among ruminants [127]. Additionally, only a few cases of chronic hepatitis E caused by zoonotic HEV-4 have been reported, while nearly all cases of chronic HEV infection were reported in patients infected with HEV-3 [128].

Regarding safety measures to lower the risk of HEV transmission via milk and food products, the efficacy of high hydrostatic pressure (HHP) processing on the infectivity of HEV has also been evaluated in pork pâté [129] and human milk [130]. In the latter, the authors applied a high pressure (600MPa, 5 min, 20 °C) to achieve significant (but not complete) inactivation of HEV in milk. While HHP treatment appeared to be more efficient than the pasteurization procedure (62.5 °C, 20 min), it is still not able to produce HEV-free milk. The results of these studies indicate that the efficacy of HPP treatment in the inactivation of HEV is matrix dependent and it might not be sufficient to fully mitigate the risk of HEV infection. 

In order to successfully inactivate HEV and decrease foodborne outbreaks, it is important to take measures for meat and milk safety, and to discourage the practice of raw liver and meat sausage consumption, as well as that of raw milk. 

## 5. Hepatitis E Virus Cross-Species Transmission

Currently, there is no clear consensus in the published literature relating to HEV in domestic ruminants. For example, goats cannot be experimentally infected with HEV-1, HEV-3, or HEV-4 [90]. In contrast, the experimental infection of sheep with a human HEV strain with an unknown genotype signified that some ruminants can be infected with human HEV [131]. Tonbak et al. also found rat HEV (*Rocahepevirus ratti*) in cattle [73]. The zoonotic HEV-3 (subtypes 3a, 3c) and HEV-4 (subtypes 4d, 4h) are the main genotypes discovered in ruminants, along with the presence of *Rocahepevirus* [97]. Rats could serve as natural HEV reservoirs, due to their continual interaction with domesticated livestock. Therefore, it is crucial to keep track of the prevalence of both human-HEV and HEV-C1 in ruminant and rat populations [132].

To minimize the potential spread of infectious diseases, farmers from EU countries are obligated to use growing chambers that prevent direct contact with rodents, other mammals, and raptors. However, the common practices to manage mixed farms, especially smaller farms in rural areas, most likely do not align very well with this requirement. Industrialized farming greatly decreases the chances for direct and indirect contact between pigs and cows compared to those in mixed or small underdeveloped farms. Interestingly, the fact that HEV is also found in cow’s milk in Turkey, where pigs are not traditionally raised, suggests the existence of other ways of infecting farmed ruminants. It is possible that domestic ruminants contract HEV infection from pastures and watering areas contaminated with the excrement of wild animals (wild boar, deer, rabbits, and rats). A nucleotide identity of 99.7% was found in full-length genome sequences of HEV strains from wild boar and deer, suggesting that interspecies transmission of HEV occurs in nature, though the mechanism of this transmission is still unclear [133]. 

Molecular analysis and epidemiological data on HEV circulation are essential tools for monitoring the virus’ spread and for designing effective prophylactic measures.

## 6. Conclusions

Various observational studies have described the presence of zoonotic HEV-3 and 4 in small and large domestic ruminants, as well as in their byproducts, demonstrating susceptibility to HEV infection and, therefore, zoonotic potential. Compared to domestic pigs, which are the primary reservoir of HEV-3 and 4 and a major source of foodborne infection in humans, domestic ruminants’ role in zoonotic spread appears to be much more limited. However, since domestic ruminants play a major part in the global food system via providing meat and milk, bettering our understanding of HEV epidemiology and ecology in livestock and the proper identification of the risk factors associated with possible transmission is vital. This requires a critical assessment of the currently inconsistent data on HEV in domestic ruminants, with an emphasis on the methodologies of detection, particularly viral load quantification in different food matrices, especially milk, and monitoring the circulation of the virus in cattle and its prevalence in traditional farms with mixed animals, as well as in semi-industrialized and industrialized farms. The duration of viremia and/or virus shedding is important but at present remains unknown; animal experiments detailing the presence of HEV in different tissues such as muscle, liver, and spleen, as well as in milk, are needed. Experimental data may also be useful in establishing the range of susceptibility and maintenance of the virus among calves and adult cows, in order to establish if they are a true reservoir of infection rather than spillover species representing a marginal risk to humans. 

## Figures and Tables

**Figure 1 viruses-16-00684-f001:**
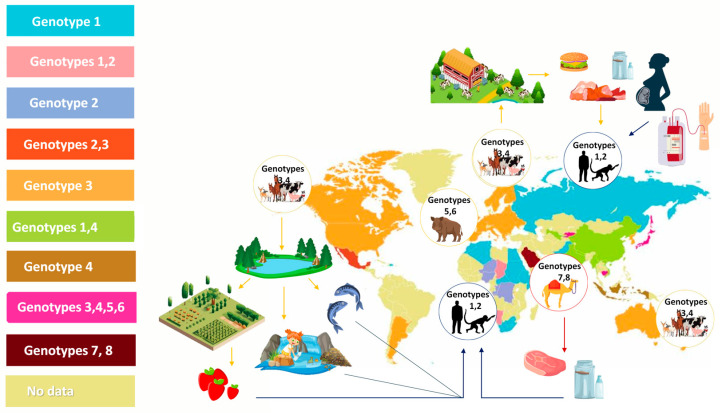
*Paslahepevirus balayani* genotypes geographical distribution and route of transmission. Different colours on the map indicate the distribution of HEV genotypes. The figure was created using data from Pérez-Gracia M.T. et al. [38].

**Table 1 viruses-16-00684-t001:** HEV RNA in goats, sheep, and cattle.

Animal	Sample Type	RNA Detection	Genotype (gt)	Reference
Goat	Faeces	9.2% (11/119)	*P. balayani* gt3	[101]
	Faeces Serum	74.1% (40/54)	*P. balayani* gt4	[62]
		60.0% (12/20)		
		53.5% (15/28)		
	Liver	4.0% (2/50)	*P. balayani* gt4	[59]
	Liver			
Sheep	Faeces	10.4% (20/192)	*P. balayani* gt3	[81]
	Serum	1.6% (3/192)		
	Liver	5.3% (4/75)	*P. balayani* gt4	[52]
	Liver	5% (3/60)	No data	[102]
	Faeces	2% (4/200)		
	Faeces	11.1% (6/54)	*P. balayani* gt4	[103]
Cow	Faeces	37.14% (52/140)	*P. balayani* gt4h	[53]
	Faeces	7. 9% (8/91)	*P. balayani* gt4	[104]
	Liver	1.0% (1/100)	*P. balayani* gt4	[105]
	Faeces	2.04% (1/45)	No data	[106]
	Faeces	14.29% (1/7)	*P. balayani* 3f	[61]
Yellow cattle (*Bos taurus*)	Serum	3% (8/254)	*P. balayani* gt4d	[107]
Yak (*Bos grunniens*)	Faeces	3.26% (3/92)	*P. balayani* gt4/3	[63]

**Table 2 viruses-16-00684-t002:** HEV RNA in detected in milk.

Animal	Sample Type and Location of Sampling	RNA Detection	Viral Load IU/mL	Genotype (gt)	References
Cow	Bulk milk from industrial farms, Germany	0% (0/400)	-	No data	[121]
Cow	Bulk Milk from industrial farms, Belgium and the Netherlands	0% (0/504)	-	No data	[122]
Cow	Milk from traditional small farms, Turkey	29.2% (14/48)	-	*P. balayani* gt1, gt3, gt4	[98]
Cow	Milk from traditional farming animals, China	37% (52/140)	-	*P. balayani* gt4h	[53]
Cow	Milk from nonmixed dairy farms, Egypt	0.2% (1/480)	10^3^ IU/mL	*P. balayani* gt3a	[118,123]
Buffalo	Milk from traditional farming animals, China	7.5% (3/40)	-	*P. balayani* gt4	[117]
Goat	Milk from mixed dairy farms, China	100% (4/4)	10^4^ to 10^5^ IU/mL	*P. balayani* gt4h	[62]
Goat	Milk from traditional farming animals in villages, Egypt	0.7% (2/280) HEV Ag (1.8%)	-	*P. balayani* gt3	[113]
Goat	Milk from traditional farming animals, Turkey	18.46% (12/65)		*P. balayani* gt1, gt3, gt4	[98]
Goat and Sheep	938 sheep and 2674 goat milk samples were pooled into 290 samples, Czech Republic	2.9% (8/290)	10^1^ to 10^3^ IU/mL	No data	[114]
Sheep	Milk from mixed dairy farms, Turkey	12.3% (8/65)	-	*P. balayani* gt1, gt3, gt4	[98]

## Data Availability

Not applicable.

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
