# Peer review of "Hepatitis E Virus in Domestic Ruminants and Virus Excretion in Milk—A Potential Source of Zoonotic HEV Infection"

_viruses, 2024, doi:10.3390/v16050684_

Round 1

Reviewer 1 Report

Comments and Suggestions for Authors

This review discusses the risk of hepatitis E virus (HEV) transmission through milk. First, the authors review the seroprevalence of HEV in domestic ruminants (sheep, goats and cattle) across the globe. Even if there is wide variation in reported HEV seroprevalence, the authors conclude that HEV (or an HEV-related virus) is well-established in domesticated ruminants. Second, the authors synthetized the data concerning the detection of HEV RNA in different samples (livers, serum, faeces, etc.) of cattle, sheep and goat also highlighting the numerous studies that did not detect HEV RNA in these animal samples. Third, a focus is made in table 2 on HEV RNA detection in milk. In the conclusion paragraph, the authors highlight the pieces of information that are lacking to better delineate HEV circulation from wild animals to ruminants and from ruminants to human beings. They also raise the question of whether these animals are spillover species or true reservoir of infection.

Recently, a systematic review and meta-analysis about hepatitis E virus detection in farmed ruminants has been published (Santos-Silva et al., 2023, Pathogens). The present review by Zhamanova et al. is more exhaustive, synthetizes data on HEV seroprevalence in ruminants and focuses on papers reporting HEV detection in milk collected from domesticated ruminants.

Introduction

Additional citations should be added to point out the zoonotic potential of HEV transmission

-       Grange et al., 2021, PNAS : HEV ranks 6th out of 887 viruses in terms of risk of transmission from animals to human.

-       Berezowski et al., 2023, EFSA J : The European Food and Safety Authority has recently classified hepatitis E on the list of the 10 priority zoonoses to be kept under surveillance

Line 35 : Reference [6] on HEV and pregnancy should be replaced by a more recent one

Lines 38-40 : the reference Pavio et al., Veterinary Research, 2017 should be added here (even if this paper has been cited later on in the manuscript)

Page 2, Figure 1.

Multiple colors are employed on Figure 1. Some are quite difficult to distinguish (ex: Genotype 2,3 versus Genotype 7,8). Try colors with better contrast. The light-yellow color has not been assigned to any legend. Maybe, it represents countries where HEV distribution is not available. This should be stated in the legend.

The word Genotype should be plural when several genotypes are mentioned.

Title of Figure 1 : “rout” with a final “e”.

The pictogram designing the transmission to non-human primates is not easily readable. The reviewer suggests to simplify the picture by showing only one individual.

Page 4 : The table dedicated to HEV detection in cattle, goat and sheep samples should be numbered 1.

In terms of the organization of the manuscript, the first table should be organized in the same order as the following paragraphs. That is : data concerning goats should appear first in the table and data concerning cattle should rank last in the table.

Paragraph 4, page 8, line 295. About the safety measures to lower the risk of HEV transmission via the milk, the high hydrostatic pressure processing has also been evaluated on the infectivity of HEV in PBS (Johne et al., 2021, Front. Microbiol.), in pork pâté (Nasheri et al., 2020) and in human milk (Bouquet et al., 2023). In the latter, the authors had to apply high pressure (600MPa, 5 min., 20°C) to achieve significant (but not complete) inactivation of  HEV in milk. This treatment appeared more efficient than the actual pasteurization procedure (62.5°C, 20 min.) that does not significantly inactivate HEV in donor milk.

This point may be discussed alongside with the temperature-based safety measures applied to milk and dairy products.

Author Response

Dear reviewer,

Thank you very much for your evaluation of our work and for your comments that will help us to improve our manuscript. The quotations recommended by you have been placed in the appropriate places, and all remarks have been taken into account. We have also worked on improving Figure 1. taking into account your comments. Table 1 was presented in the order you recommended. Further, we discussed milk safety and HHP processing. We include the text:

Regarding the safety measures to lower the risk of HEV transmission via milk and food products, the efficacy of high hydrostatic pressure (HHP) processing on the infectivity of HEV has also been evaluated in pork pâté [123] and human milk [124]. In the latter, the authors applied high pressure (600MPa, 5 min., 20°C) to achieve significant (but not complete) inactivation of  HEV in milk. While HHP treatment appeared more efficient than the pasteurization procedure (62.5°C, 20 min.), it is still not able to produce HEV-free milk. The results of these studies indicate that the efficacy of HPP treatment in the inactivation of HEV is matrix-dependent, and it might not be sufficient to fully mitigate the risk of HEV infection. 

Reviewer 2 Report

Comments and Suggestions for Authors

To the editor of Viruses (MDPI):

The Critical Analysis entitled “Hepatitis E Virus in domestic ruminants and virus excretion in 2 milk – a potential source of zoonotic HEV infection” by Gergana Zahmanova, Katerina Takova, Georgi L. Lukov and Anton Andonov, is a well written and interesting paper. The authors investigated the current knowledge in literature about the circulation of HEV in ruminants and raw milk.

They should pay attention to insert a brief “methods” paragraph in which they describe how do they collected bibliographic information, indicating what was used to screen information (internet tools used, specific key words, which kinds of papers were accepted or discharged … why… etc.) and if work was produced to test the robustness of gained information level.

However, this manuscript could be accepted, should the authors consider major revisions and minor comments to improve the quality of the work.

English language is well adopted.

Major revisions:

LINE 36: the authors should specify that these infections are mainly due to non-zoonotic HEV gen 1 and gen 2 (Asia, Africa…)

LINE 122-123: how is possible that cattle HEV seroprevalence in Europe reached 29.35% if the maximum value was found in Bulgaria (7.7%) (by a non-cited manuscript, perhaps Tsachev et al., 2023)?

LINE 145-148: Confusing: if the authors are writing about molecular detection of HEV RNA (e.g. by PCR), they cannot justify a cross-reactive HEV immune response by HEV-like particles, because this is evident by immunological methods. Maybe this works better: ‘… due to infection with HEV-like viruses which are genetically quite different (and not detectable by PCR), but still inducing…’

Are there only two works that hypothesize the existence of these HEV-like particles?

The authors should decide to use “fecal (feces)” or “faecal (faeces)” throughout all the text

LINE 162 and 164: isn’t it Table 1?

Table 1 and LINE 176: data on goat serum are not clear (15/28 positive sera = 53.6%). In fact, the reference number 52 (Long et al., 2016) described 5/28 positive sera! Am I wrong?

Minor revisions:

LINE 35-36: ‘…come from low income areas, where…’

LINE 40: ‘…fact that HEV human seroprevalence has…’

LINE 47: ‘…additional ORF4, whose translation stimulates viral…’

LINE 58: ‘…while the eHEV particles circulate in the blood and, in vitro, in infected cell-culture supernatants…’

LINE 59-60: sentence not clear: it seems that both eHEV and neHEV are similarly infectious

LINE 66-74: the authors do not describe HEV gen 5, 6, and 8.

Figure 1: please, cite the source of this image even if it is an own production.

LINE 96: any information by the authors about avian and bat HEV?

LINE 100: Can the authors explain if milk contamination happens only during the viraemic phase or also in other moments of the infection?

LINE 99: ‘… domesticated ruminants has risen…’

LINE 100: ‘… and especially concerning the excretion of HEV in milk …’

LINE 103: ‘…of previous works that found HEV in…’

LINE 107-109: ‘…Most ruminants belong to the bovids family (Bovidae), within which the subfamily Bovinae includes cattle and related Yaks and buffalo among others, and the subfamily Caprinae covers 20 species including domestic goats and sheep…’

LINE 120: ‘… HEV seroprevalence, the detected types of immuno-globulins, …’

LINE 123: the authors should specify what Abs means and, more important, they should cite the work from Bulgaria

LINE 123: insert ‘if it is excluded’ instead of ‘unlike’ Spain

LINE 123: ‘… in the Americas data widely vary: the USA …’

LINE 127: ‘… (21.6%) [65], except for South Korea …’

LINE 131: ‘… In the Americas there has been only one …’

LINE 128-139: harmonize using only simple past

LINE 138: ‘… (0-100%); in Africa: Egypt …’

LINE 141: ‘… seroprevalence, it seems …’

LINE 145: host

LINE 151-52: ‘…basis from birth to weeks-old goats, HEV RNA detection was never detected and, at the same time, anti-HEV seroconversion has been observed [79].’

LINE 155: ‘…from ruminants. Silva et al. determined that…’

LINE 172:’…  genotype 3c highly matching …’

LINE 187: ‘… living together with the owners …’

LINE 193: ‘… with no molecular evidence of HEV infection …’ What do the authors mean? Only ELISA detection or only PCR detection without genotyping?

LINE 201: ‘… Mongolia; however, the detection rate was higher in liver samples (3/60, or 5%) [95].’

LINE 205: ‘… and from the Czech Republic in milk samples (4/290, 1.4%) [103].’

LINE 203: ‘… As was the case with goats above, no HEV RNA was found in equal number of studies in fecal, serum and/or liver specimens from China, India, Turkey and Spain…’

LINE 212: ‘… from 0.19% (3/1692), to 0.8% (7/912) in cows and 3.15% (8/25) in yellow cattle…’ It is not clear what these percentages are referred to (HEV RNA or Antibodies in sera?)

LINE 213: ‘… 37.14%...’ Confusing again: milk OR feces?

LINE 231: ‘…Advancing further westward…’

LINE 245: ‘…spread and activating prophylactic measures.’

LINE 251: summarizes

LINE 259: 18-month-old

LINE 277: ‘…including pigs, carry a higher risk…’

LINE 278: ‘…transmission than in highly industrialized…’

LINE 281: ‘…always properly followed in small, family-run farms…’

LINE 289: ‘… it is possible to hypothesize environmental contamination of the milk due to poor operators hygienic practices …’

LINE 290-293: ‘Further to that … … HEV infection.’ The authors should better explain the meaning of this sentence.

LINE 300: ‘…56°C has still intact HEV… … when pigs are inoculated…’

LINE 301: ‘…studies shows that HEV cannot be inactivated by briefly heating at 56°C or 60°C…’

LINE 311-12: ‘… HEV and decrease foodborne outbreaks… … and discourage the practice of raw liver...’

LINE 327-28: ‘…the common practices to manage mixed farms, especially smaller farms in rural areas, most likely do not align very well…’

Comments on the Quality of English Language

minor revisions

Author Response

REVIEWER 2

Dear reviewer,

Thank you very much for your evaluation of our work and for your comments that will help us improve our manuscript.

Major revisions:

LINE 36: the authors should specify that these infections are mainly due to non-zoonotic HEV gen 1 and gen 2 (Asia, Africa…)

Answer: Thanks for your comment. We changed the text: The majority of the patients with HEV infections come from areas with few resources (Asia and Africa), where the infection is transmitted mainly by the fecal-oral route and due to the non-zoonotic HEV genotypes 1 and 2 (HEV-1 and HEV-2).

They should pay attention to insert a brief “methods” paragraph in which they describe how do they collected bibliographic information, indicating what was used to screen information (internet tools used, specific key words, which kinds of papers were accepted or discharged … why… etc.) and if work was produced to test the robustness of gained information level.

Answer: We do not claim to present a systematic review, which requires a certain approach in the search and selection of articles covering keywords. For this reason, we believe that it is not necessary to include a new chapter that would have described the methodology of searching and selecting the articles we have cited.

LINE 122-123: how is possible that cattle HEV seroprevalence in Europe reached 29.35% if the maximum value was found in Bulgaria (7.7%) (by a non-cited manuscript, perhaps Tsachev et al., 2023)?

Answer: The sentence was rewritten according to your comments. Worldwide the HEV seroprevalence in cows varies from 0.0% to 29.35. Tsachev et al., 2023 was cited.

LINE 145-148: Confusing: if the authors are writing about molecular detection of HEV RNA (e.g. by PCR), they cannot justify a cross-reactive HEV immune response by HEV-like particles, because this is evident by immunological methods. Maybe this works better: ‘… due to infection with HEV-like viruses which are genetically quite different (and not detectable by PCR), but still inducing…’

Answer: The sentence was rewritten according to your comments.  As mentioned above this may be due to infection with HEV-like viruses which are genetically quite different (and not detectable by PCR), but still inducing cross-reactive HEV immune response. Indeed, an HEV-like moose virus which was highly genetically divergent has been identified in Sweden.

Are there only two works that hypothesize the existence of these HEV-like particles?

Answer: We believe that these two works provide enough information to support our hypothesis.

The authors should decide to use “fecal (feces)” or “faecal (faeces)” throughout all the text

Answer: We used “faecal (faeces)” throughout the text.

LINE 162 and 164: isn’t it Table 1?

Yes. We numbered it: Table 1.

Table 1 and LINE 176: data on goat serum are not clear (15/28 positive sera = 53.6%). In fact, the reference number 52 (Long et al., 2016) described 5/28 positive sera! Am I wrong?

Answer: We believe that we have quoted correctly Long et al., 2016: Moreover, viremia was found in 53.57% (15/28) goats.

Minor revisions:

LINE 35-36: ‘…come from low income areas, where…’ Answer: Your correction was included in the text.

LINE 40: ‘…fact that HEV human seroprevalence has…’ Answer: Your correction was included in the text.

LINE 47: ‘…additional ORF4, whose translation stimulates viral…’ HEV genotype 1 has an additional ORF4, its translation stimulates viral replication [14] Answer: Your correction was included in the text.

LINE 58: ‘…while the eHEV particles circulate in the blood and, in vitro, in infected cell-culture supernatants…’ Answer: Your correction was included in the text.

LINE 59-60: sentence not clear: it seems that both eHEV and neHEV are similarly infectious

The sentence was rewritten: The enveloped forms of the virus are somewhat more impenetrable to anti-HEV antibody, because the host membrane completely masks the viral antigens [25]. eHEV is responsible for cell-to-cell spread within the host [26]. neHEV is characterized by a high initial cell attachment (∼10-fold higher) than that of the enveloped virions [27]. 

LINE 66-74: the authors do not describe HEV gen 5, 6, and 8.

Answer: HEV-5 and HEV-6 were observed among wild boars []. HEV-7 infects mostly Dromedary camels, however its zoonotic potential has been demonstrated as well, albeit in a single human case (an immunosuppressed patient from the Middle East, who consumed camel meat and milk) [24,25]. HEV-8 was identified among Bactrian camel in China

Figure 1: please, cite the source of this image even if it is an own production. Answer: Your correction was included in the text.

LINE 96: any information by the authors about avian and bat HEV?

Answer: Dear reviewer, thank you very much for your valuable comment. For the purposes of our review, we consider it unnecessary to present the complete classification of the subfamily Orthohepevirinae. We focused on  Paslahepevirus balayani and Rocahepevirus ratti, that can infect people. Also, if you think it is necessary, we can include the following information in the text: Avian HEV strains belong to the genus Avihepevirus and lead to liver and spleen disease in birds. Bat HEV strains belong to the genus Chirohepevirus and has no evidence of zoonotic potential.

LINE 100: Can the authors explain if milk contamination happens only during the viraemic phase or also in other moments of the infection? Answer: Contamination of the milk happens during the viraemic phase.

LINE 99: ‘… domesticated ruminants has risen…’ Answer: Your correction was included in the text.

LINE 100: ‘… and especially concerning the excretion of HEV in milk …’ Answer: Your correction was included in the text.

Answer: Your correction was included in the text.

Recently a growing awareness of the zoonotic potential of HEV in domesticated ruminants has risen, and especially concerning the excretion of HEV in milk and the implications of a possible milk-borne transmission

LINE 103: ‘…of previous works that found HEV in…’

Answer: Your correction was included in the text.

LINE 107-109: ‘…Most ruminants belong to the bovids family (Bovidae), within which the subfamily Bovinae includes cattle and related Yaks and buffalo among others, and the subfamily Caprinae covers 20 species including domestic goats and sheep…’

Answer: Your correction was included in the text.

LINE 120: ‘… HEV seroprevalence, the detected types of immuno-globulins, …’

Answer: Your correction was included in the text.

LINE 123: the authors should specify what Abs means and, more important, they should cite the work from Bulgaria

Answer: Your correction was included in the text.

LINE 123: insert ‘if it is excluded’ instead of ‘unlike’ Spain

Answer: Your correction was included in the text.

LINE 123: ‘… in the Americas data widely vary: the USA …’

Answer: Your correction was included in the text.

LINE 127: ‘… (21.6%) [65], except for South Korea …’

Answer: Your correction was included in the text.

LINE 131: ‘… In the Americas there has been only one …’

Answer: Your correction was included in the text.

LINE 128-139: harmonize using only simple past Answer: Your correction was included in the text.

LINE 138: ‘… (0-100%); in Africa: Egypt …’ Answer: Your correction was included in the text.

LINE 141: ‘… seroprevalence, it seems …’ ‘… Answer: Your correction was included in the text.

LINE 145: host Answer: Your correction was included in the text.

LINE 151-52: ‘…basis from birth to weeks-old goats, HEV RNA detection was never detected and, at the same time, anti-HEV seroconversion has been observed [79].’ Answer: Your correction was included in the text.

LINE 155: ‘…from ruminants. Silva et al. determined that…’ Answer: Your correction was included in the text.

LINE 172:’…  genotype 3c highly matching …’ Answer: Your correction was included in the text.

LINE 187: ‘… living together with the owners …’ Answer: Your correction was included in the text.

LINE 193: ‘… with no molecular evidence of HEV infection …’ What do the authors mean? Only ELISA detection or only PCR detection without genotyping?

Answer: Under the absence of molecular evidence we consider the absence of proven HEV RNA.

The sentence was changed: It is important to mention that there is a growing list of published reports from diverse geographical regions with no molecular evidence of HEV infection (no HEV RNA detection) when testing of the most relevant specimens such as serum, bile and faeces was performed.

LINE 201: ‘… Mongolia; however, the detection rate was higher in liver samples (3/60, or 5%) [95].’ Answer: Your correction was included in the text.

LINE 205: ‘… and from the Czech Republic in milk samples (4/290, 1.4%) [103].’ Answer: Your correction was included in the text.

LINE 203: ‘… As was the case with goats above, no HEV RNA was found in equal number of studies in fecal, serum and/or liver specimens from China, India, Turkey and Spain…’ Answer: Your correction was included in the text.

LINE 212: ‘… from 0.19% (3/1692), to 0.8% (7/912) in cows and 3.15% (8/25) in yellow cattle…’ It is not clear what these percentages are referred to (HEV RNA or Antibodies in sera?)

Answer: Thanks for your comment, these percentages are referred to HEV RNA. We rewrite the sentence.

LINE 213: ‘… 37.14%...’ Confusing again: milk OR feces?

Answer: 140 faecal samples - 37.14%

LINE 231: ‘…Advancing further westward…’ Answer: Your correction was included in the text.

LINE 245: ‘…spread and activating prophylactic measures.’ Answer: Your correction was included in the text.

LINE 251: summarizes Answer: Your correction was included in the text.

LINE 259: 18-month-old Answer: Your correction was included in the text.

LINE 277: ‘…including pigs, carry a higher risk…’ Answer: Your correction was included in the text.

LINE 278: ‘…transmission than in highly industrialized…’Done

LINE 281: ‘…always properly followed in small, family-run farms…’Done

LINE 289: ‘… it is possible to hypothesize environmental contamination of the milk due to poor operators hygienic practices …’ Done

LINE 290-293: ‘Further to that … … HEV infection.’ The authors should better explain the meaning of this sentence.

LINE 300: ‘…56°C has still intact HEV… … when pigs are inoculated…’Done

LINE 301: ‘…studies shows that HEV cannot be inactivated by briefly heating at 56°C or 60°C…’ Done

LINE 311-12: ‘… HEV and decrease foodborne outbreaks… … and discourage the practice of raw liver...’ Done

LINE 327-28: ‘…the common practices to manage mixed farms, especially smaller farms in rural areas, most likely do not align very well…’ Done

DOIhttps://doi.org/10.1128/jvi.02804-15
